# Health Service Interventions for Intimate Partner Violence among Military Personnel and Veterans: A Framework and Scoping Review

**DOI:** 10.3390/ijerph19063551

**Published:** 2022-03-17

**Authors:** Sean Cowlishaw, Alyssa Sbisa, Isabella Freijah, Dzenana Kartal, Ashlee Mulligan, MaryAnn Notarianni, Katherine Iverson, Anne-Laure Couineau, David Forbes, Meaghan O’Donnell, Andrea Phelps, Patrick Smith, Fardous Hosseiny

**Affiliations:** 1Phoenix Australia–Centre for Post-traumatic Mental Health, Department of Psychiatry, The University of Melbourne, Level 3, Alan Gilbert Building, 161 Barry Street, Carlton, VIC 3053, Australia; alyssa.sbisa@unimelb.edu.au (A.S.); isabella.freijah@unimelb.edu.au (I.F.); dkartal@unimelb.edu.au (D.K.); annelc@unimelb.edu.au (A.-L.C.); dforbes@unimelb.edu.au (D.F.); mod@unimelb.edu.au (M.O.); ajphelps@unimelb.edu.au (A.P.); 2Centre of Excellence on Post-Traumatic Stress Disorder and Related Mental Health Conditions, 1145 Carling Avenue, Ottawa, ON K1Z 7K4, Canada; ashlee.mulligan@theroyal.ca (A.M.); maryann.notarianni@theroyal.ca (M.N.); patrick.smith@theroyal.ca (P.S.); fardous.hosseiny@theroyal.ca (F.H.); 3Women’s Health Sciences Division of the National Center for PTSD, Veterans Affairs Boston Healthcare System, 150 South Huntington Street, Boston, MA 02130, USA; katherine.iverson@va.gov; 4Department of Psychiatry, Boston University School of Medicine, 72 E Concord St, Boston, MA 02118, USA

**Keywords:** intimate partner violence, veteran, military, intervention, health services

## Abstract

IPV is a significant concern among active duty (AD) military personnel or veterans, and there is a need for initiatives to address violence perpetrated by such personnel, and IPV victimisation in military and veteran-specific contexts. The aim of this paper was to provide an overview of major IPV intervention approaches and evidence in military and veteran-specific health services. A scoping review was conducted involving a systematic search of all available published studies describing IPV interventions in military and veteran-specific health services. Findings were synthesised narratively, and in relation to a conceptual framework that distinguishes across prevention, response, and recovery-oriented strategies. The search identified 19 studies, all from the U.S., and only three comprised randomised trials. Initiatives addressed both IPV perpetration and victimisation, with varied interventions targeting the latter, including training programs, case identification and risk assessment strategies, and psychosocial interventions. Most initiatives were classified as responses to IPV, with one example of indicated prevention. The findings highlight an important role for specific health services in addressing IPV among AD personnel and veterans, and signal intervention components that should be considered. The limited amount of empirical evidence indicates that benefits of interventions remain unclear, and highlights the need for targeted research.

## 1. Introduction

Intimate partner violence (IPV) can describe any behaviour in a current or former intimate relationship that results in physical, psychological, or sexual harm [1]. This may involve acts of physical and sexual aggression, or psychological forms of abuse including coercive and controlling behaviours that aim to dominate the victim and restrict their autonomy [2]. Although IPV can be directed towards men, these exposures have particularly heavy impacts on women and gender or sexual minorities. By way of illustration, IPV is the leading cause of disease burden for women aged 25–44 years in some countries [3], while the majority of homicides among women across jurisdictions are also perpetrated by intimate partners [4]. IPV has many further non-fatal impacts on victims, resulting in physical conditions and injuries, as well as mental health problems including depression and post-traumatic stress disorder (PTSD) [5,6].

IPV is a significant concern among active duty (AD) military personnel or veterans, with available studies indicating high rates of violence perpetrated by current or former personnel [7], and alarming levels of IPV exposure (or victimisation) across relevant populations [8]. By way of illustration, a recent systematic review of population-based surveys and population screening studies indicated approximately one in five of all AD personnel and veterans reported recent (e.g., past year) exposures to IPV, with analogous figures of approximately one in eight reported for any IPV perpetration [9]. This review also examined sources of variability across populations and settings, and indicated comparable levels of IPV perpetration and victimisation among men and women, and elevated rates of perpetration among veterans, relative to AD personnel. Finally, the review suggested higher rates of IPV perpetration in studies of military or veteran-specific health services, relative to general samples of personnel, along with a contrasting trend towards lower rates of victimisation identified in such specific services.

The high rates of IPV among AD personnel and veterans suggests a pressing need for initiatives to address violence in military and veteran-specific contexts. Relevant initiatives may include programs that focus on preventing violent behaviour, holding perpetrators accountable, and enhancing safety among victims, and have been considered mainly in civilian contexts. However, military occupational health models highlight important features of this employment context (e.g., military cultural values, separate health and judicial systems) which mean that civilian modes of intervention can have variable effects and benefit from adaptation to account for occupational and cultural considerations in military or veteran-specific environments [10]. In this context, the aim of this paper was to provide an overview of major approaches to addressing IPV and associated evidence, via a scoping review of studies of relevant interventions in military or veteran-specific health services. This focus also draws attention to a distinct body of research addressing IPV in such specific service use environments, and aligns with literature indicating that health services have central roles in multi-sectoral responses to IPV and violence against women more generally [11].

## 2. Materials and Methods

### 2.1. Identifying Relevant Studies

A scoping review was conducted [12] involving a systematic search of studies of IPV interventions in military and veteran-specific health services. Searches of electronic databases (PsycINFO, CINHAL, PubMed, and the Cochrane Library) combined terms relating to IPV, military populations, health settings, and interventions, and were conducted on 26 May 2020. See Appendix A for terms. Records were screened on title and abstract by one reviewer, and on full text by two reviewers.

### 2.2. Study Selection

Eligible studies comprised any design and described an intervention targeting IPV perpetrators or victims in military or veteran-specific health services; the latter defined by provision of specific care for physical or psychiatric issues of AD personnel or veterans. These could be from any country, and examples included Veteran Affairs (VA) Medical Centres in the United States, and Operational Stress Injury Clinics in Canada. Eligible studies included pilot studies or trials of interventions, and descriptive studies or evaluations of implementation processes.

Intervention studies were included if they were delivered in specific health services, or if participants were recruited entirely or substantially from these settings. Eligible studies also reported on IPV identification strategies that were considered in such services, including self-report tools or clinician-administered measures. Given the heterogeneity of studies describing IPV measures among patients recruited from relevant services, the search was restricted to studies reporting data regarding utility, performance, or implementation of measures. This included studies addressing questions regarding acceptability, accuracy, and psychometric properties of measures, as well as implementation processes. Thus, the search excluded studies that only described results regarding prevalence or correlates of IPV. Additionally excluded were papers that did not present any empirical data from patients regarding relevant interventions, with the exception of protocol papers for ongoing trials.

### 2.3. Data Charting

Data were charted regarding study characteristics including authors, year of publication, intervention type, target population, setting, and key findings. No appraisal of evidence quality was conducted, consistent with the scoping review methodology [12].

### 2.4. Summarising and Reporting Findings

A narrative synthesis of studies was also conducted, in accordance with a proposed conceptual framework for IPV interventions. This was based on the Institute of Medicine (IOM) continuum of services [13], but was adapted to account for heterogeneous policies and programs that have been proposed to address IPV perpetration and victimisation, respectively. This framework initially distinguished across three types of prevention strategy: universal prevention, which targeted entire populations, assuming average levels of overall risk for IPV; selective prevention strategies targeting specific sub-populations exhibiting elevated risk; and indicated prevention strategies which target individuals displaying early signs of IPV use or exposure, but before these have progressed to severe levels of threat or impact. The framework also identified an additional category of responses to IPV, which focused on individuals who were currently experiencing or using violence. Finally, the framework identified further interventions defined by a subsequent focus on recovery; for example, that addressed long-term mental health or psychosocial problems following IPV exposure.

## 3. Results

### 3.1. Search Results

The PRISMA flowchart is displayed in Figure 1. From a yield of 2162 records, 1585 were screened on title and abstract, and 173 on full text. Studies were excluded for various reasons (e.g., interventions not situated in health services). There were 19 eligible studies, and all originated from the U.S. Ten studies identified interventions targeting IPV perpetration, ten targeting victimisation, and one addressed IPV victimisation and perpetration. Three studies involved randomised designs. Characteristics of studies are described in Appendix A, while Appendix A provides additional details of randomised trials.

Table 1 provides an overview of interventions that were described in eligible studies. As shown, there were multiple studies addressing responses to IPV victimisation, including training programs for health providers (k = 2), case identification strategies (k = 5), and brief or extended psychosocial interventions (k = 2). There were multiple studies addressing responses to IPV perpetration, which all comprised IPV treatment programs that were group based (k = 4) or were classified as ‘integrative’ or ‘eclectic’ programs (k = 2). There was one example of a couples-based indicated prevention strategy, while there were no instances of universal or selective prevention strategies, and no recovery-oriented interventions. 

### 3.2. Narrative Synthesis

#### 3.2.1. IPV Perpetration

The search identified two studies describing one program that was classified as an indicated prevention strategy that comprised a group intervention for couples called the Strength at Home—Couples (SAH-C) program [15]. Given the orientation to work with couples, SAH-C could be classified under prevention programs for both IPV perpetration and victimisation in Table 1. However, this intervention is described in the current section only.

SAH-C comprises a 10 week trauma-informed group program that targets couples comprising veterans (who are men) and their partners that report relationship distress or psychological aggression, but no physical violence perpetration by men, and no ‘severe’ physical violence (that produced fear or injury) by women. SAH-C is based on a cognitive behavioural model which assumes that trauma can produce biases in social information processing that increases risk of violent behaviours. Modules focus on understanding impacts of trauma on relationships, conflict management, and communication skills training [15]. 

Published reports regarding SAH-C summarise findings from one pilot study [14] and one randomised trial [15] involving male AD personnel or veterans, and their female partners, recruited partly from veteran-specific health services. It was required that men did not endorse severe IPV perpetration based on the Revised Conflict Tactics Scale (CTS-2) [14]. The pilot study involved sequential allocation of couples (*n* < 10 treatment completers) to two conditions (SAH-C versus Supportive Therapy) and provided descriptive data regarding veteran reports and partner reports of IPV across conditions. This pilot also required that personnel or veterans met criteria for PTSD, although this criterion was relaxed in the randomised trial. The latter involved random allocation of *n* = 69 couples to SAH-C versus Supportive Therapy, and post-treatment comparisons suggested small to moderate benefits of SAH-C in terms of reducing physical and psychological IPV [15] (see Appendix A). However, most effects were not statistically significant (*p* < 0.05), and trends should be viewed as provisional given the small sample for a prevention trial, and thus low levels of statistical power.

The search identified four studies that addressed a related program classified as a response strategy for IPV perpetrators. The Strength at Home—Men program (SAH-M) for AD personnel or veterans was intended to end use of IPV in current relationships and prevent future violence, and was based on the same model underlying SAH-C [14,15] that emphasised social information processing biases linked to trauma exposure and PTSD. SAH-M is a 12 week group program involving weekly sessions, psychoeducation, group activities, and practice assignments. Content focuses on psychoeducation, readiness to change, conflict and anger management, and improving coping and communication skills.

Published reports on SAH-M include two pilot studies [16,31], one randomised trial [17], and one report on evaluation data from an implementation pilot involving SAH-M in select VA hospitals [32]. The pilot studies involved small samples (*n* < 10) and no control conditions. One pilot also reported findings from focus groups regarding acceptability [31]. A subsequent randomised trial compared SAH-M to enhanced treatment as usual (ETAU) [17] (see Appendix A). AD personnel or veterans were eligible if they reported physical IPV perpetration or ongoing IPV-related legal issues. Findings indicated that SAH-M was associated with reductions in reports of IPV perpetration across post-treatment and 3 month follow-up, when compared to ETAU, and effects were most apparent at immediate post-treatment. Finally, the report of evaluation data from an implementation pilot at 10 VA hospitals considered uptake and engagement with SAH-M across the first year of the program, and reported declines in IPV pre- to post-intervention [32].

Two studies described alternative programs that were also classified as response strategies for IPV perpetrators. One comprised an individual program for IPV and co-occurring substance use problems, which included a Motivational Interviewing (MI) session targeting physical violence, followed by 5 sessions of CBT [18]. These addressed anger arousal and conflict escalation, conflict management, problem solving, and communication difficulties. An extended version included 12 weeks of Continuing Care (CC) involving telephone monitoring. A randomised trial compared these versions (MI-CBT and MC-CBT+CC) with ETAU among primarily male veterans seeking substance use or mental health treatment [18]. Results indicated that MC-CBT+CC demonstrated benefits over time relative to ETAU (see Appendix A). 

The final intervention comprised a 13 week Batterer Intervention Program for male veterans that was administered through the VA Medical Centre Domestic Relations Clinic [19,20]. There was limited detail reported regarding content, except that the program was cognitive behavioural and recovery-oriented and adopted a non-confrontational approach. The program was described as eclectic and including group and individual sessions, as well as mandatory psychoeducation classes and case management. While this program was administered through a VA clinic, the participants also included veterans referred from outside of health services and were often mandated to attend. Published reports described sample characteristics and treatment engagement, as well as pre–post outcomes. 

#### 3.2.2. IPV Victimisation

Table 1 identifies interventions that were categorised as responses to IPV victimisation. These included training for service providers in identification and responses to IPV, case identification and danger assessment strategies, and brief or extended psychosocial interventions.

Two studies described training for military health providers regarding IPV victimisation. The first comprised a training program for military dental practitioners termed PANDA (Prevent Abuse and Neglect through Dental Awareness) [21]. The 1 h training aimed to enhance capacities of practitioners to identify signs of IPV, including head and neck injuries, document injuries, and onward refer suspected cases [21]. The published report described surveys before and after implementation, and focused mainly on awareness of regulations and protocols regarding abuse and neglect. The second program [22] comprised a 7 h training for providers who were largely behavioural health specialists. Training involved handouts, presentations, case scenarios, and guidelines regarding referrals and crimes related to IPV, and roleplay. There was limited detail otherwise about content, except that curriculum emphasised overlaps involving IPV and substance abuse, suicide, murder-suicide, as well as referrals and crimes related to IPV. The report considered data from pre-training and post-training surveys, and addressed change in knowledge, attitudes, and efficacy regarding IPV. 

There were five studies describing elements of case identification strategies in military and veteran-specific health services, encompassing tools and protocols for improving recognition of service users who have experienced IPV. Four studies described performance or implementation of brief IPV screening tools, including the 4-item Hurt-Insult-Threatens-Scream (HITS) [23,24], and an extended version called the E-HITS [25,26]. Published reports addressed accuracy and sensitivity when using the CTS-2 as the reference standard among women veterans. One publication also reported data supporting feasibility and acceptability [24]. The implementation of routine screening using the E-HITS in VHA primary care clinics is currently under evaluation in an ongoing implementation-effectiveness trial [26]. The search also returned one study which reported on the 10-item Trauma Questionnaire, which includes 2 items about threatened and actual IPV [27]. This considered data from patients attending a VA women’s health centre, and considered sensitivity and specificity when compared to clinicians screening for trauma. 

In the context of case identification, a 3-item version of the Danger Assessment scale has also been considered as a secondary IPV risk measure [28]. This comprises questions from the longer Danger Assessment [33] that asks about (a) increases in violence, (b) strangulation, and (c) victims’ belief about threat to life. These provide indications of current danger levels, including risk of severe violence and intimate partner homicide. The report on implementation in VA medical centres suggested that utilisation could facilitate increased engagement with psychosocial services, although adoption of the secondary screener was modest at the time of reporting [28].

There were two studies describing psychosocial interventions in military and veteran-specific health services. The first comprised a modular computer-based intervention called Safe and Healthy Experiences (SHE) [29]. SHE has been situated in a VA women’s primary care clinic, and involved electronic screening of patients for lifetime sexual trauma, IPV, hazardous drinking, and PTSD [29]. Women screening positive received a module involving psychoeducation resources and assessments of readiness to change, which determined provision of resource handouts or feedback to address ambivalence about change. At the time of review, there is one published report on SHE, comprising an open trial (no control condition) involving *n* = 20 women veterans, and *n* = 10 completed the IPV module [29]. 

The second program comprised a patient-centred and trauma-informed counselling intervention that was developed in veteran-specific health services called Recovering from IPV through Strength and Empowerment (RISE) [30]. RISE is flexible in length and content and comprises 1 to 6 sessions addressing topics which women veterans select. These include safety planning, IPV health effects, coping and self-care, enhancing social support, making difficult decisions, and connecting with resources. RISE incorporates MI and targets self-efficacy, patient activation, empowerment, and distress. Published reports comprise a protocol for an open trial and subsequent RCT to determine effectiveness for improving psychosocial functioning of women veterans who have recently experienced IPV [30].

## 4. Discussion

This scoping review addressed IPV intervention approaches that have been considered in published studies of military and veteran-specific health services. This identified a predominant focus on response-oriented strategies for IPV, including significant examples of responses to both IPV perpetration and victimisation. However, the findings also highlight a broader range of initiatives addressing IPV victimisation, relative to perpetration, including training for health providers, case identification and risk assessment strategies, and select forms of psychosocial intervention. In contrast, there was a comparably narrow range of approaches to IPV perpetration, comprising group treatments and programs for men, with no comparable studies of training for service providers or case identification strategies. There was one example of an intervention for couples that reported relationship distress or psychological aggression, but no physical violence perpetrated by men, which by virtue of this focus on both partners was classified as an indicated prevention strategy. The review otherwise identified no interventions that reflected universal or selective prevention strategies, and no approaches were classified as promoting recovery from effects of IPV.

The findings should be viewed in the context of few studies involving randomised designs, and thus limited evidence regarding benefits of interventions in relevant contexts. Accordingly, it cannot be assumed that approaches are associated with benefits that outweigh harms, and also justify costs, and these questions remain critical considerations. The findings should also be viewed in relation to the specific focus of the search, which targeted studies based in military and veteran-specific health services, and thus omitted evidence situated outside of such environments. For example, this would exclude the Family Advocacy Program in the U.S. [34,35], which provides domestic violence services that are not necessarily embedded in health care settings, as well as screening protocols for IPV perpetration that are at pre-implementation stages of development [36]. Despite the fact that the search was not inclusive of all evidence, the current focus aligns with non-military literature that has emphasised the critical role of health services in addressing IPV [11]. The findings suggest that this extends to military and veteran-specific health services, which may thus provide equivalent contexts for responses to IPV among AD personnel and veterans. The review also signals that IPV prevention, in contrast, is likely to require involvement of other agencies and departments that have contact and responsibility for all AD personnel or veterans (not just those accessing services), and non-military agencies also involved in violence prevention more broadly. 

Notwithstanding the narrow scope of the search, the absence of studies describing some intervention approaches, including varied responses to IPV perpetrators, and recovery-oriented interventions for victims, is likely to reflect meaningful gaps that suggest areas of need for investment and innovation. These may parallel areas identified in non-military settings, including via prior reviews that have acknowledged scant evidence for interventions targeting IPV perpetrators in health environments generally [37]. Comprehensive reviews of psychological interventions for IPV victims have also identified that most are situated in crisis settings (e.g., domestic violence shelters) [38], with few examples of trials of therapies for women who are no longer in unsafe situations that would be classified as ‘recovery-oriented’ interventions. Such gaps seem particularly important in veteran-specific health services, given high rates of IPV perpetration and victimisation among service users, and thus expected levels of unmet need across populations. 

There were several other important trends that were discernible in eligible studies. While studies from any country were potentially eligible, the review indicated that all identified reports were situated in the U.S., and accordingly it highlights the absence of empirical attention to IPV interventions for AD personnel and veterans in other jurisdictions. Furthermore, there was a modest number of reports overall (k = 19), and these addressed different interventions using varied methodologies. Critically, there were only three interventions considered in randomised trials. Two were based on the SAH framework [15,17], which emphasises social information processing biases associated with trauma, and does not strongly address power and control dynamics that are targets of many programs for coercive and controlling IPV [39]. Despite this, the randomised trial of SAH-M suggested beneficial effects [17], while SAH-C also demonstrated promise (although findings were equivocal given low statistical power) [15]. These trends may highlight value from interventions for IPV perpetrators that are adapted to accommodate features of military occupational and environmental contexts. 

### Limitations

The search considered studies situated in military or veteran-specific health services, generally involving service users in heterosexual relationships, and it did not identify intervention approaches or evidence considered outside of such contexts, and with heterogeneous populations. Second, the eligibility criteria were purposively broad and included diverse studies providing varying standards of evidence. Despite the broad criteria, as with all systematic searches, it is possible that studies were missed in the search. Furthermore, the review did not involve any risk of bias assessment, while the pool of eligible studies was small and identified few randomised trials. As such, the review should be viewed as indicating possible interventions that could have promise in military and veteran-specific services but require research to demonstrate that these produce beneficial effects. Finally, while the proposed framework provides a way of organising approaches to IPV that may have utility in military and veteran-specific health services, it remains likely that the model will not fully capture the heterogeneity and nuance of all possible interventions and may need to be elaborated in future research.

## 5. Conclusions

IPV is a complex psychosocial problem, and it should be anticipated that meaningful reductions among AD personnel and veterans will require comprehensive strategies that involve prevention, response, and recovery-oriented interventions, along with cross-sector cooperation of military and veteran-specific agencies, and non-military organisations. The review focuses attention on military and veteran-specific health services, and thus highlights an important context for initiatives comprising responses to IPV. Such services may provide logical places to focus efforts within jurisdictions that are still at early stages of developing strategies for addressing IPV. Multiple interventions for IPV victimisation could be considered, including training programs for service providers on how to recognise and respond to victims, protocols for improving case identification and risk assessment, along with psychosocial support programs that provide referral targets for service users who disclose IPV. In addition, the review highlights the importance of interventions that support AD personnel and veterans who disclose IPV perpetration and wish to change their behaviour, along with the need for new resources that may include training for service providers on how to recognise and respond to IPV perpetration, as well as tailored case identification or risk assessment strategies. Although this review indicates that evidence underlying these approaches remains limited, the growing recognition of high rates of IPV among AD personnel and veterans, along with potentially lethal consequences for victims, strongly indicates that this should not preclude attempts to develop safe and sensible responses to IPV in these specific services.

The limitations of available evidence signal ongoing needs for research to guide decisions about what works and how interventions can be improved in military and veteran-specific health services. This includes a specific need for intervention studies outside the U.S., and across jurisdictions including Canada, Australia, and the UK. These all have different systems for providing care for AD personnel and veterans, as well as family members, which may have implications for feasibility and effects of interventions. Second, there is also a need for studies to inform selection of interventions targeting IPV perpetration among AD personnel and veterans. These may include new and enhanced ways of supporting perpetrators to change their behaviour, evaluations of benefits of training programs for service providers, as well as studies of case identification strategies. Third, there is also a need for research on different components of intervention approaches for IPV victims. This includes sophisticated studies involving randomised designs that can demonstrate benefits of training programs (for example, see Lotzin et al. [40]) and case identification strategies (see MacMillan et al. [41]).

Finally, there is also a need for studies considering interventions comprising responses to IPV, as well as recovery-oriented interventions for victims. The former may include low-intensity counselling interventions delivered by military and veteran-specific health providers, or para-professionals, as well as intensive programs delivered by specialist ‘advocates’ embedded in relevant services, or within community-based agencies. The latter may include trauma-focused treatments (for example, see Kubany et al. [42]) and other treatments for distress [43] or PTSD [44], that could help promote recovery from any long-term mental health and psychosocial sequela of IPV. However, the range of recovery-oriented interventions that may be beneficial should also be viewed broadly, and future studies should consider initiatives involving wrap-around services, including but not limited to provision of employment support [45,46] and long-term housing [47].

## Figures and Tables

**Figure 1 ijerph-19-03551-f001:**
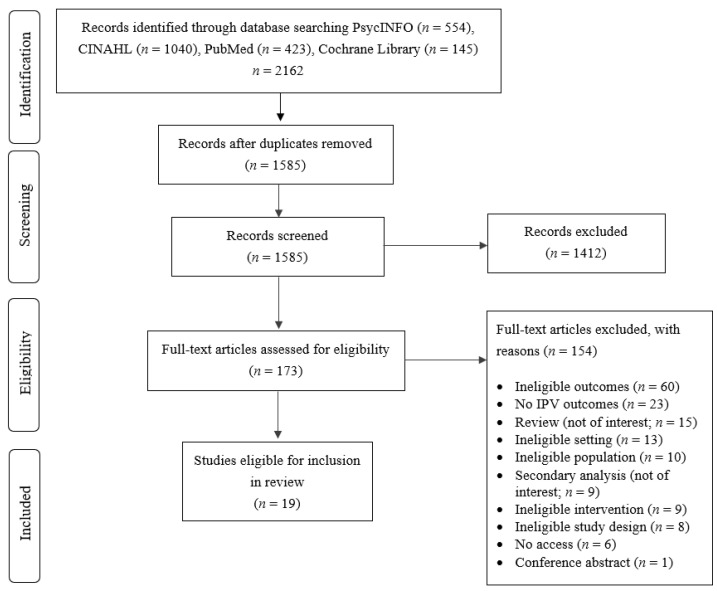
PRISMA flow diagram.

**Table 1 ijerph-19-03551-t001:** Summary of IPV prevention, response and recovery interventions from published literature.

Prevention			Response	Recovery
Universal	Selective	Indicated		
Perpetrator focused				
NA	NA	**Couple programs**	**Group treatment programs**	NA
		SAH-C program [14,15]	SAH-M program [16,17] **Integrative/eclectic programs**	
			MI, CBT, and telephone monitoring for physical violence and co-occurring substance use [18] VA Domestic Relations Clinic—Batterer Intervention Program [19,20]	
Victim focused				
NA	NA	**Couple programs**	**Training programs for service providers**	NA
		SAH-C program [14,15]	Prevent Abuse and Neglect through Dental Awareness (PANDA) [21] Instructional Curriculum for VA care providers [22] **Case identification strategies**	
			4-item HITS screening tool [23,24] 5-item E-HITS screening tool [25,26] 2 items from the Trauma Questionnaire [27] **Risk assessment strategies** 3-item Danger Assessment [28] **Brief psychosocial intervention** Safe and Healthy Experiences (SHE) module [29] **Extended psychosocial interventions** Recovering from IPV through Strength and Empowerment (RISE) [30]	

Notes: NA = none available; SAH-C = Strength at Home—Couples; SAH-M = Strength at Home—Men; MI = Motivational Interviewing; CBT = cognitive behavioural therapy; VA = Veteran Affairs; HITS = Hurt-Insult-Threatens-Scream; E-HITS = Extended Hurt-Insult-Threatens-Scream.

## Data Availability

Not applicable.

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
