# Peer review of "Health Service Interventions for Intimate Partner Violence among Military Personnel and Veterans: A Framework and Scoping Review"

_ijerph, 2022, doi:10.3390/ijerph19063551_

Round 1

Reviewer 1 Report

This is probably the best written and soundest methods paper I've reviewed for the last 20 or so papers.  I did not find a single technical error, which is very unusual. 

It is not clear why the Family Advocacy Program (line 337) was excluded as I thought it did deal with military families, at least through a counseling approach, which is certainly a health-related approach at the least.  This should be explained more clearly and the readers should be informed of relevant resources/citations/references if they wished to explore IPV work/research along those lines.

Author Response

The review was focussed on intervention approaches that were situated in military and veteran-specific health services, which for current purposes were defined by the provision of specific care for physical or psychiatric issues of AD personnel or veterans. In contrast, the Family Advocacy Program comprises a social service intervention which is focussed directly on addressing domestic violence, child abuse and neglect, as well as problematic sexual behaviour in children and youth. These services are typically located on military installations, and are independent of specific health services. If there had been a report on a component of the Family Advocacy Program that was embedded in a military-specific health service, then this would have been included in the review. However, our search identified no such studies that could be included on this basis.

We believe that the focus on interventions in specific health services is defensible and aligns with non-military literature that has also focussed on IPV interventions in health service environments including primary care and psychiatric services. However, we also recognise that this narrow scope is a limitation and have acknowledged this in the body of the discussion  and the limitations. In response to this comment, however, we have made further insertions in the discussion to better explain the reasoning for excluding this program, and have also provided a link to an additional citation about the Family Advocacy Program (line 420-422).

Reviewer 2 Report

The main aim of the article is clear and relevant. The introductory section provides a suitable framework; I would just add a comment:

When referring to the incidence of IPV [line 60], an article by almost the same authors is cited, but under review, so it cannot be consulted. Instead, other articles can be cited, such as:

- J. Kwan, K. Sparrow, E. Facer-Irwin, G. Thandi, N.T. Fear, D. MacManus. Prevalence of intimate partner violence perpetration among military populations: A systematic review and meta-analysis. Aggression and Violent Behavior, 2020, Vol 53, 101419. https://doi.org/10.1016/j.avb.2020.101419.

- K. Sparrow, H. Dickson, J. Kwan, L. Howard, N. Fear, D. MacManus. Prevalence of Self-Reported IntimatePartner Violence Victimization Among Military Personnel: A Systematic Review and Meta-Analysis. Trauma, Violence and Abuse, 2018, I-24. https://doi.org/10.1177/1524838018782206.

The materials and methods section is well planned and obtains relevant results, correctly describing the analyzed articles.

The discussion section shows the difficulty of addressing strategies against IPV among military and veterans in a context of few published studies and limited to health services.

Author Response

As suggested, we have inserted references to the recommended articles (line 56-57).

Reviewer 3 Report

Thank you for the chance to review your scoping review of health service interventions for IPV. As I was reading your article, I had more thoughts related to the articles that ended up being reviewed than your own. I think you did a nice job with explaining the process in which you identified the articles chosen for review as well as the results you found within these articles. I agree with your assessment that this review further supports the gap that is needing filling for military personnel and veterans who both experience or perpetrate IPV.

The one thing I think would help with clarification is indicating if you did or did not include articles from any country. You provided the examples from the US and Canada and then referenced Canada, Australia, and the UK in your conclusion. I actually flipped back to your methods while reading your conclusion to get the clarification of whether studies from all these countries were reviewed or not.

Author Response

While studies from any country were eligible for the review, our search only identified reports on interventions from the U.S. We have made insertions to make this more clear in the method (line 99) and discussion (line 460).